# Detection and Prevention of Medication Errors by the Network of Sentinel Pharmacies in a Southern European Region

**DOI:** 10.3390/jcm12010194

**Published:** 2022-12-27

**Authors:** Anna M. Jambrina, Àlex Santomà, Andrea Rocher, Neus Rams, Glòria Cereza, Pilar Rius, Montserrat Gironès, Clara Pareja, Àngels Franch, Manel Rabanal

**Affiliations:** 1Directorate-General for Healthcare Planning and Regulation, Ministry of Health, Government of Catalonia, 08028 Barcelona, Spain; 2Physiology Section, Department of Biochemistry and Physiology, Faculty of Pharmacy and Food Science, University of Barcelona, 08028 Barcelona, Spain; 3Department of Pharmacology, Therapeutics and Toxicology, Universitat Autònoma de Barcelona, 08193 Bellaterra, Spain; 4Council of the Pharmacist’s Association of Catalonia, 08009 Barcelona, Spain; 5Nutrition and Food Safety Research Institute (INSA-UB), 08921 Santa Coloma de Gramenet, Spain

**Keywords:** adverse drug reaction (ADR), medication error, medication safety, drug-related problems, community pharmacies, health services administration, pharmacoepidemiology

## Abstract

A medication error (ME) is a drug-related problem that has been recognized as a common and serious threat to patient safety. The aim of this study was to detect and analyze ME reports occurring throughout the therapeutic process through the community’s pharmacies in order to improve the efficacy and safety of medications and contribute to the prevention of future MEs. This was a three-year descriptive, observational, and prospective study to detect and analyze the different MEs reported by the Catalan sentinel pharmacies network (Catalan SePhaNet). In total, 1394 notifications of MEs were reported (an incidence rate of 737.34 cases/100,000 inhabitants). MEs were detected more frequently in primary care centers. Most of the MEs reported were caused by an incorrect, incomplete, illegible, or verbal medical prescription (41.3%). Of the global notifications detected, 71.9% did not reach the patient (categories A and B). The drugs most frequently implicated in the reported ME cases were beta-lactam antibiotics. In 6.0% of the cases, the ME caused injury to the patient (categories E and F). In 72.0% of the global notifications, a pharmacist’s intervention avoided the ME. The importance of a community pharmacy and the role of a pharmacist were demonstrated in aspects related to patient and drug safety.

## 1. Introduction

A medication error (ME) is defined as any preventable event that may cause or lead to inappropriate medication use or patient harm while the medication is in control of the health care professional or patient [1].

MEs, together with adverse drug reactions (ADRs), are drug-related problems that have been recognized as common and serious threats to patient safety [2]. In this sense, drug-related problems are important causes of morbidity and mortality, as well as hospital admissions and lengthened patient stays, thereby increasing costs for the health care system [3].

An ME occurs when weak medication systems and/or human factors affect prescribing, transcribing, dispensing, administering, and monitoring practices [4]. MEs can cause undesirable adverse drug events, and in some cases, they can lead to life-threatening injuries [5]. Reducing preventable harm in healthcare, as well as the errors that lead to them, has long been recognized as a patient safety priority; recent reports estimate that 5% of patients are exposed to preventable harm during their medical care [6,7].

As an important but underappreciated part of the healthcare system, community pharmacies play a vital role in patient safety by ensuring that medications are used safely by patients [8]. At the same time, community pharmacies are the closest health care centers for patients because of their accessibility and geographical capillarity, and they are where pharmacists can take care of most of the low-complexity processes.

To this end, in Catalonia, a region of Southern Europe, a network of sentinel pharmacies (Catalan SePhaNet) has been created and promoted [9]. This network is a new sentinel surveillance model in pharmacy offices that has been integrated into health systems where a community pharmacist is able to participate in shared management scenarios. In this context, community pharmacists collaborate with health administration in the observation, detection, and reporting of events of interest related to the surveillance of medicine prescription, dispensing, and administration, as well as in the description of population-level behaviors regarding important health problems. The Catalan SePhaNet works in several lines of performance; one of them is in the detection, notification, and pharmacist intervention in cases of suspected ME. The network pharmacies run these activities in order to strengthen the existing voluntary notification system and verify if the actions in terms of prevention of MEs are useful, as well as to increase drug safety [9].

In this sense, community pharmacists can help improve health by acting in certain aspects. On the one hand, they can decrease the probability of ADRs and increase the reporting of low-severity events detected during dispensing. On the other hand, they can promote greater adherence to medication, strengthening and improving the quality of health care.

In this study, we detected and analyzed ME reports occurring in the therapeutic process through the Catalan SePhaNet in order to improve the efficacy and safety of medications and contribute to the prevention of future MEs.

## 2. Materials and Methods

We conducted a three-year descriptive, observational, and prospective study (January 2019–December 2021) in order to detect and analyze the different MEs reported by Catalan SePhaNet community pharmacies in Catalonia, Spain.

### 2.1. Sampling Frame

The Catalan SePhaNet, formed by 75 community pharmacies scattered throughout the region, was selected proportionally to a stratification of the population of Catalonia based on criteria of representativeness, ensuring a coverage of approximately 2.5% of the total population of Catalonia being monitored [9]. 

The inclusion criteria were a patient who attended at the community pharmacy with a foreseeable incident that could result in inappropriate use of the medication or harm [1]. Dual cases—cases involving incomplete information and relating to products which were not drugs—were excluded.

The community pharmacist only detected MEs at the time of dispensing the medication during their daily care practice in the pharmacy, without having access to the patient’s medical records. The ADRs corresponding to the severe ME cases were defined by the European Medicine Agency (EMA) as noxious and unintended responses to medicine [10]. The ADRs were reported directly to the Regional Pharmacovigilance Centre. The responsible technicians of this Centre recorded the case, analyzed the data, and accessed the patient’s medical history, as required. The community pharmacists were only in charge of detecting the cases, and the evaluations of causality were carried out by professionals from the Regional Pharmacovigilance Centre.

Participant pharmacists were trained by expert pharmacists from the Catalan Ministry of Health and the Council of the Pharmacist’s Association of Catalonia. The workshop had a duration of 4 h and was developed in two parts. Initially, a theoretical part was taught to improve the knowledge of the pharmacists in these subjects. Next, real cases of received notifications were used, with the aim of improving the information collected and the ability to detect cases and acquire the knowledge and skills necessary to carry out the activity. The main topics of this training were the theoretical bases of MEs, their severity, the causes, the places of detection, and the patient harms associated with the ADRs. Several training sessions were performed to maintain a regular exchange of information and resolve questions.

### 2.2. Data Collection

We gathered the data through an anonymous 11-item electronic form filled in by the community pharmacists during their daily care practice whenever a ME was detected. The electronic form filled in was a validated questionnaire (“Notification of medication error”) and was grouped in four different parts: pharmacy identification, ME classification, medication involved in the ME, and pharmacist management. The recorded variables were:Code and name of sentinel pharmacyDate of ME detectionClassification of the ME severity (choice between severity categories A to I) according to the Institute for Safe Medication Practices (ISMP) classification [11]National code of medicinal productName of medicinal productME origin (choice between pharmacy office, primary care center, hospital, patient residence, nursing home, and other)Cause of the ME (multiple answer: prescription, verification, dispensing, administration, similarity of packaging, similar names, incorrect labelling, incorrect dose, incorrect preparation, lack of information, system errors, non-compliance of the patient, therapeutic duplicity, and others)ME has been avoided (yes or no)Description of the preventive pharmaceutical action performedIf the ME had caused an injury to the patient and a description of the associated ADR typeObservations (free text field to indicate any relevant aspect during the ME detection)

Patient information was obtained anonymously by observation during the interview and neither verbal nor written consents were needed. All confidential information collected was recorded in the applications portal of the Ministry of Health, accessible by username and password through the Drugs and Pharmacy Channel website. Overall, it allowed us to obtain information about the type of ME that occurred most frequently and the involvement of the pharmacist in their prevention.

### 2.3. Statistical Analysis

The variables were categorical and were summarized as numbers of (notified) cases and percentages. We grouped the numbers of notifications by quarters for the statistical analysis. Then we used the Kruskall–Wallis test to find the significant differences between the groups. For the multiple comparison, we applied a Dunn’s test post hoc. For the purposes of this study, we considered a *p*-value of <0.05 to be statistically significant. All the analyses were conducted with the XLSTAT software (Data Analysis and Statistical Solution for Microsoft Excel, Addinsoft, Paris, France 2017) [12].

## 3. Results

### 3.1. Incidence Data

During the study period, 1394 notifications of MEs were reported by Catalan SePhaNet. This represents a three-year cumulative incidence rate of 737.34 cases/100,000 inhabitants. The incidence of ME notifications during the pre-pandemic phase of the SARS-CoV-2 period in 2019 overall (302.05 cases/100,000 inhabitants) was lower than the incidence rate during the pandemic period in 2020 and 2021 (435.30 cases/100,000 inhabitants). No significant differences were observed overall. 

A more detailed incidence data analysis per quarter (Figure 1) showed differences for the second quarter of 2020 (corresponding to the pandemic lock-down) compared to that of 2019 (*p* < 0.05). Likewise, there was a difference between the fourth quarter of 2021 (corresponding to the omicron variant outbreak) and the same quarter of 2019 (*p* < 0.05).

### 3.2. Typology of Medication Error

Figure 2 shows the reported MEs by places of detection. We detected MEs more frequently in primary care centers, followed by the patient’s home, hospital, other sites, pharmacy office, and nursing homes. No statistically significant differences were observed throughout the three-year study between the places where MEs occurred.

Most of the MEs reported were caused by an incorrect, incomplete, illegible, or verbal medical prescription (41.3%), followed by 12.4% of the cases being due to an incorrect prescribed dosage (Table 1). The total number of cases exceeded the 1394 notifications received because, in most cases, the ME was caused by a different reason at the same time. Throughout the three-year study period, the most frequent causes that produced the MEs followed the same trend. No statistically significant differences were observed.

Regarding ME severity, as shown in Table 2, 71.9% of the global notifications detected did not reach the patient (categories A and B) and 28.0% of them affected the patient (categories C–I), while 6.0% of the MEs resulted in injuries, some of which were serious (categories E and F). No critical or catastrophic cases of MEs were reported (categories G–I). The data from 2021 showed an increasing trend of severe reported ME notifications.

### 3.3. Drugs Involved

The Anatomical Therapeutic Chemical (ATC) Classification System ranks the active ingredients of drugs according to the organ or system on which they act, as well as their therapeutic, pharmacological, and chemical properties [13].

Table 3 shows that according to the organ or system in which the drug acts, the drugs most frequently implicated in the reported MEs were those related to the nervous system (19.1%), followed by drugs that affected the cardiovascular system (15.9%), drugs that act on the alimentary tract and metabolism (15.3%), and lastly, anti-infectives for systemic use (11.8%). The total number of cases exceeded the 1394 notifications received since, in most cases, there was more than one drug implicated in the ME.

Similarly, the drugs most frequently implicated in the reported ME cases were beta-lactam antibiotics, followed by antithrombotic agents, anti-inflammatory and antirheumatic products (non-steroids), antidepressants, Vitamins A and D (including combinations of the two), and other analgesics and antipyretics (Table 4).

In 6.0% of the total reported cases, the ME caused injury to the patient (categories E and F), and 0.6% of those were serious (category F). Table 5 describes the nine serious reported cases during the study period, which means that in these cases, the patients required hospitalization or hospitalization had been prolonged and the ME caused temporary injury.

### 3.4. Pharmacovigilance and Pharmaceutical Care

The Catalan SePhaNet detected 6.0% of the MEs associated with ADRs. These ADRs corresponded to the severe cases that injured the patients (categories E and F) and were reported to the Regional Pharmacovigilance Centre.

In 72.0% of the cases, the pharmacist’s intervention avoided the ME either by pharmacotherapeutic monitoring (whether at the time of dispensing or during the preparation of pillboxes), not dispensing the medication, referring the patient to the doctor or consulting directly with him, or substituting the medication (depending on the type of ME). In some notifications, the ME was detected through the computer program. In other cases, the patients reported them directly.

In 73.7% of the cases, additional pharmaceutical actions were taken by carrying out a control and/or reviewing the medication, educating the patient regarding the rational use of the medication, confirming with the prescriber the correct regimen, and training the staff who worked in the pharmacy office, separating misleading medication, among others.

## 4. Discussion

Patient safety is a key pillar of healthcare quality. The growing interest in this area has largely stimulated research to measure and report on the organizational attributes believed to influence patient safety. One such attribute is safety culture, defined as a product of individual and group values, attitudes, perceptions, competencies, patterns of behavior that determine the commitment to safety culture, and the style and proficiency of an organization’s health and safety management [14].

Understanding the safety culture of community pharmacies can greatly contribute to organizational quality improvement efforts by raising staff awareness about patient safety issues, as well as by identifying the areas of strength and those that require improvement [8].

The Catalan SePhaNet has contributed to the effectiveness and efficiency of traditional surveillance programs by providing valid data related to drug safety. Furthermore, it has allowed the identification of individual and population-level behaviors and provided an overall view about the burden of disease and the risks associated with the use of medication [15,16,17].

Through the network of sentinel pharmacies, we are able to detect, notify, analyze, and prevent several MEs detected during the dispensing process, as described in the currently available literature [5,7,18,19,20,21,22,23,24]. Complementarily, MEs detected in the hospital and primary care settings primarily affect the processes of the therapeutic chain related to prescription and administration [25,26,27].

The data we obtained during the study period (January 2019–December 2021) demonstrated an effect of the COVID-19 pandemic on the incidence of reported MEs. A statistically significant decrease in incidence rate was observed during the second quarter of 2020 compared to the same period in 2019, which corresponded to the pandemic lock-down beginning in March 2020. Similarly, during the fourth quarter of 2021, the decrease in the incidence rate of MEs compared to the same period in 2019 was due to the increase in cases because of the omicron variant outbreak. Current studies demonstrating an effect of the COVID-19 pandemic’s decrease in ME cases focus on the primary care and hospital settings and are limited with respect to community pharmacies [28]. On the other hand, some studies have shown an increase in MEs, primarily in the elderly, due to the chronicity of their diseases, polypharmacy, and functional deterioration, which are aggravated by staying at home [29].

It is interesting to note that approximately half of the cases of MEs detected by community pharmacies in Catalonia occurred in primary care centers. In contrast, in other countries such as France, pharmaceutical interventions to prevent MEs related to hospital medical orders were higher [20]. These differences are primarily related to the type of communication between professionals from different health fields, which are necessary to implement simple measures such as shared medical records or direct electronic communication between professionals.

The analysis of ME causes shows that several factors intervene due to the fact that an error can occur at any point in the therapeutic chain (prescription, validation, dispensing, and administration). The data suggest that an incorrect, incomplete, illegible, or verbal medical prescription, followed by an incorrect dose, were the primary causes of MEs detected in community pharmacies in Catalonia. This has also been observed by other international studies that analyzed in depth the MEs detected during the dispensing process, as well as quality-related events [5,18,19,22,30,31,32,33].

Regarding the severity of MEs, in most cases, the community pharmacist was able to prevent errors (71.9%). Only a small percentage (6.0%) caused ADR-related damage to patients, and these were reported to the Regional Pharmacovigilance Centre to assess causality. This trend is similar to that observed in other studies from Canada and Norway that looked at a higher number of MEs and highlighted the importance of preventing prescription errors and adverse events associated with medications [32,34]. This further highlights the importance of the role of the community pharmacist as the link in the health chain in preventing safety incidents and contributing to the rational use of medicine.

The ME description is as important as the drug involved in an error. There is currently sparse literature available that collects this information; therefore, this study offers complete descriptive information on the type of ME detected in a community pharmacy. The drugs most frequently implicated in reported ME cases were beta-lactam antibiotics (5.0%), followed by antithrombotic agents (4.6%). These findings are similar to those of a study conducted in Australia, where systemic antibiotics was the therapeutic group most involved in MEs [30].

Finally, our study also demonstrated the importance of pharmaceutical intervention at the time of dispensing, which contributes to avoiding a significant number of MEs (70%) through pharmaceutical prevention actions (73.7%), in the same way that other European studies have shown [23]. Currently, there are more comprehensive studies focusing on the type of pharmaceutical action. They describe interventions triggered by patient-reported problems with prescribed medicines [35], as well as the types of corrective actions and prevention strategies recommended to improve patient safety [31].

## 5. Conclusions

The Catalan SePhaNet demonstrates the importance of the community pharmacy and the role of the pharmacist in aspects related to patient and drug safety. Specifically, this study allowed us to analyze in detail the MEs detected during the dispensing process, providing significant information. It has been demonstrated that a community pharmacist can avoid most MEs, and they interfered on those errors that caused an injury to the patient. The most serious MEs associated with ADRs should be influenced, with the implementation of improvement strategies focused on enhanced surveillance and multidisciplinary communication. Spanish law requires pharmacists to counsel patients, and research shows that counselling can assist with detecting MEs. Community pharmacies play an important role in the health system, especially in processes of low complexity; even so, their efforts to reduce MEs have had short visibility. The diffusion of these results and the promotion of pharmacist involvement to improve patient and drug safety provides the community pharmacy with an active role in reducing MEs and safeguarding patients from harm.

## Figures and Tables

**Figure 1 jcm-12-00194-f001:**
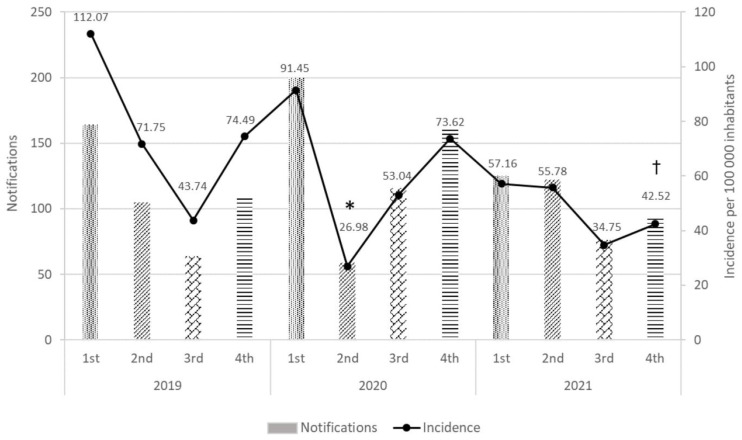
Number of ME notifications and incidence rates per quarter (Catalonia, 2019–2021). Significant differences in incidence: * vs. second quarter of 2019, † vs. fourth quarter of 2019 (*p* < 0.05).

**Figure 2 jcm-12-00194-f002:**
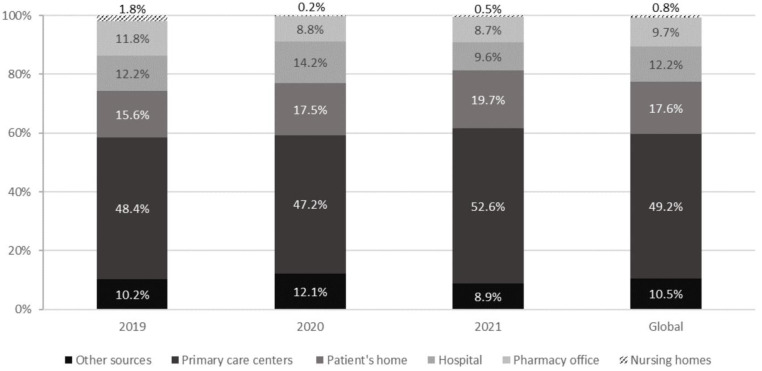
Percentages of reported MEs based on the places where the MEs occurred (Catalonia, 2019–2021).

**Table 1 jcm-12-00194-t001:** Numbers and percentages of ME cases based on the causes and processes of the therapeutic chain involved in the ME (Catalonia, 2019–2021).

Cause of ME	Cases 2019		Cases 2020		Cases 2021		Global Cases	
	(*n* = 632)	%	(*n* = 748)	%	(*n* = 579)	%	(*n* = 1959)	%
Incorrect, incomplete, illegible, or verbal medical prescription	239	37.8	337	45.1	234	40.4	810	41.3
Incorrect prescribed dosage	97	15.3	81	10.8	65	11.2	243	12.4
Therapeutic duplication	40	9.8	58	6.8	59	5.2	157	8.0
Incorrect administration	46	7.3	55	7.4	47	8.1	148	7.6
Lack of information	62	6.3	51	7.8	30	10.2	143	7.3
Patient non-compliance	19	4.9	34	3.3	27	3.3	80	4.1
Similarity of packaging	31	3.5	25	2.5	19	3.3	75	3.8
Incorrect dispensing	22	3.2	19	3.1	19	2.8	60	3.1
Other causes	20	3.2	23	2.1	16	3.6	59	3.0
Similar names	20	3.0	16	4.5	21	4.7	57	2.9
System error (structure, process, or organization)	12	2.8	28	1.3	15	3.8	55	2.8
Incorrect prescription verification	18	1.9	10	3.7	22	2.6	50	2.6
Incorrect preparation	5	0.8	8	1.1	4	0.7	17	0.9
Incorrect or misleading labelling	1	0.2	3	0.4	1	0.2	5	0.3

**Table 2 jcm-12-00194-t002:** Numbers and percentages of ME notifications based on severity category (Catalonia, 2019–2021).

ME Severity Category	Notifications 2019		Notifications 2020		Notifications 2021		Global Notifications	
	(*n* = 442)	%	(*n* = 536)	%	(*n* = 416)	%	(*n* = 1394)	%
A: Circumstance capable of causing an ME	31	7.0	21	3.9	14	3.4	66	4.7
B: The ME occurred but was detected before reaching the patient	295	66.7	379	70.7	263	63.2	937	67.2
C: The ME did not cause injury to the patient	80	18.1	103	19.2	88	21.2	271	19.4
D: The patient required observation, but no injury occurred	10	2.3	9	1.7	17	4.1	36	2.6
E: The patient required treatment and/or the ME caused temporary injury	23	5.2	23	4.3	29	7.0	75	5.4
F: The patient required hospitalization or hospitalization has been prolonged and has caused temporary injury	3	0.7	1	0.2	5	1.2	9	0.6
G: The ME caused a permanent injury to the patient	0	0.0	0	0.0	0	0.0	0	0.0
H: The ME caused a situation that came close to causing the death of the patient	0	0.0	0	0.0	0	0.0	0	0.0
I: The ME caused or contributed to the patient’s death	0	0.0	0	0.0	0	0.0	0	0.0

**Table 3 jcm-12-00194-t003:** Numbers and percentages of drugs involved in the MEs reported according to the organ or system on which the drug acted, based on ATC code descriptions (Catalonia, 2019–2021).

ATC Code	ATC Code Description	Global Notifications	
		(*n* = 1505)	%
N	Nervous system	287	19.1
C	Cardiovascular system	239	15.9
A	Alimentary tract and metabolism	230	15.3
J	Anti-infectives for systemic use	178	11.8
R	Respiratory system	133	8.8
B	Blood and blood-forming organs	89	5.9
M	Musculoskeletal system	85	5.6
H	Systemic hormonal preparations, excluding sex hormones and insulins	72	4.8
S	Sensory organs	55	3.7
D	Dermatological	54	3.6
G	Genitourinary system and sex hormones	44	2.9
Others	Medical devices and food supplements	14	0.9
L	Antineoplastic and immunomodulating agents	13	0.9
P	Antiparasitic products, insecticides, and repellents	9	0.6
V	Various	3	0.2

**Table 4 jcm-12-00194-t004:** Numbers and percentages of the drugs most frequently involved in the MEs reported, according to the classifications by therapeutic group based on the ATC code descriptions (Catalonia, 2019–2021).

ATC Code	ATC Code Description	Global Notifications	% Global Notifications
J01C	Beta-lactam antibiotics and Penicillins	75	5.0
B01A	Antithrombotic agents	69	4.6
M01A	Anti-inflammatory and antirheumatic products and non-steroids	58	3.9
N06A	Antidepressants	58	3.9
A11C	Vitamins A and D, including combinations of the two	56	3.7
N02B	Other analgesics and antipyretics	55	3.7
A02B	Drugs for peptic ulcer and gastro-esophageal reflux disease	45	3.0
N02A	Opioids	45	3.0
A10B	Blood glucose-lowering drugs, excluding insulins	44	2.9
H02A	Corticosteroids for systemic use, plain	44	2.9
C10A	Lipid-modifying agents, plain	38	2.5
N05B	Anxiolytics	35	2.3
R03B	Other drugs for obstructive airway diseases and inhalants	34	2.3
A10A	Insulins and analogues	32	2.1
N05A	Antipsychotics	31	2.1
J01F	Macrolides, lincosamides, and streptogramins	29	1.9
C09A	Selective calcium channel blockers with direct cardiac effects	28	1.9
R06A	Antihistamines for systemic use	27	1.8
N03A	Antiepileptics	26	1.7
R03A	Adrenergics and inhalants	25	1.7
C03C	High-ceiling diuretics	25	1.7
J01X	Other antibiotics	25	1.7
A12A	Mineral supplements	24	1.6
C09D	Angiotensin II receptor blockers (ARBs) and combinations thereof	23	1.5
C07A	Beta-blocking agents	22	1.5
S01E	Antiglaucoma preparations and miotics	20	1.3
C08C	Ace inhibitors and combinations thereof	20	1.3
H03A	Thyroid preparations	20	1.3

**Table 5 jcm-12-00194-t005:** Characteristics of the serious ME cases reported (Catalonia, 2019–2021).

Drug	Source of ME	Cause of ME	Additional Pharmaceutical Action	ADR Associated
Rocoz^®^ 100 mg (API * Quetiapine)	Pharmacy office	Incorrect dispensing	The two pillboxes causing the error were separated due to similarity in the names of the patients.	Undefined
Clopidogrel Normon 75 mg EFG ^†^	Patient’s home	Lack of information and patient non-compliance	The pharmacist urged the family to review the medication with the prescribing physician.	Stroke due to patient non-compliance
Augmentine^®^ 500 mg (API * Amoxicillin/clavulanic acid)	Primary care center	Incorrect medical prescription	None	The patient attended the primary care center with chest pain angina and was prescribed Augmentine^®^ 500 mg. Two days later, he returned to the emergency room because he had chest pain. The patient was referred to the hospital, where pericarditis was detected due to an incorrect prescription of the antibiotic dose.
Fentanyl^®^ Stada 75 µg EFG ^†^	Patient’s home	Incorrect administration	The patient applied the new patch without removing the old one because he thought the drug had worn off. The pharmacist explained to the patient when and how to remove the fentanyl patch.	Drowsiness, hypotension, and light headedness that resulted in requiring treatment in the primary care center.
Furosemide Cinfa 40 mg EFG ^†^	Primary care center	Incorrect medical prescription and system error (structure, process, or organization)	None	The patient had edema in the leg because the doctor forgot to renew the furosemide in the electronic prescription and the patient stopped taking the medication. The patient was hospitalized for ten days.
Eliquis^®^ 2.5 mg (API * Apixaban)	Hospital	Incorrect medical prescription	None	The patient was taking acenocoumarol to prevent thromboembolism secondary to a mechanical heart valve. After the change in medication from acenocoumarol to apixaban, the patient suffered another myocardial infarction, with hospitalization and subsequent surgical intervention.
Sintrom^®^ 4 mg (API * Acenocoumarol)	Primary care center	Incorrect prescribed dosage and system error (structure, process, or organization)	None	Elevated international normalized ratio (INR)
Amoxicillin/clavulanic acid Mylan 500 mg EFG ^†^	Patient’s home	Other causes	Due to the ADR experienced by the patient, the pharmacist advised referring him to the doctor so that he could take it into account in future prescriptions.	After the administration of six doses of antibiotic, the patient stopped the treatment due to the appearance of hemorrhoids. The patient attended the hospital, where she was prescribed a cream for the hemorrhoids.
Prednisone Alonga 50 mg EFG ^†^	Primary care center	Incorrect medical prescription and lack of information	The patient was diabetic and suffered facial paralysis. Prednisone 50 mg was prescribed and her basal glycemia rose to 600 mg/dL. The patient was admitted to the hospital and was under observation, where she was administered rapid insulin. The pharmacist advised her to periodically check her glucose levels and take note of when the medication was stopped.	Hyperglycemia

* API, active pharmaceutical ingredient. ^†^ EFG, generic drug.

## Data Availability

The datasets that support the findings of this study are available from the first author (A.M.J.) upon reasonable written request.

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
