# Peer review of "Detection and Prevention of Medication Errors by the Network of Sentinel Pharmacies in a Southern European Region"

_jcm, 2022, doi:10.3390/jcm12010194_

Round 1
Reviewer 1 Report
The authors have chosen an important public health topic. It is a very well written article. I suggest few following minor considerations:
· Abstract: Well written do not need any revision.
· Introduction: Well written do not need any revision.
· Methodology: State inclusion/exclusion criteria for categorization of ME. Were any specific guidelines were used?
· Results and discussion; In result section, how ADR were identified? Which scale? Did you also assessed causality?

Author Response
Dear Reviewer:
First, we want to thank you for the comments and suggestions received. We have tried to incorporate all the changes that you have suggested. Next, we present the changes made.
- Methodology: State inclusion/exclusion criteria for categorization of ME. Were any specific guidelines were used?
The inclusion criteria were a patient who showed up at the community pharmacy with a foreseeable incident that could result in inappropriate use of the medication or harm. Dual cases, cases involving incomplete information and relating to products which are not medicinal products are excluded.
We have used the Institute for Safe Medication Practices (ISMP) and we have included all this information in the matherial and methods.
- Results and discussion; In result section, how ADR were identified? Which scale? Did you also assessed causality?
We have included this information in the results and discussion sections. A small percentage (6.0%) caused ADR-related damage to the patient and was reported to the Regional Pharmacovigilance Centre to assess causality.
Thank you so much for all the interesting considerations.
Best regards.
Reviewer 2 Report
First of all, congratulations to all the authors for addressing such an interesting and much needed study.
1) Title reflects the body of manuscript.
2) Introductions highlights the rationale of study, need of the study and purpose of the study. However, the research question is missing in the introduction section and the same can be incorporated.
3) Methodology needs further elaboration in order to make the things more clear and replicable.
4) Results are comprehensive.
5) Conclusion and references are precise and uniform.
The article is robust and written well. However, the quality can be improved by incorporating the specific comments.
Author Response
Dear Reviewer:
First, we want to thank you for the comments and suggestions received. We have tried to incorporate all the changes that you have suggested. Next, we present the changes made.
- Introductions highlights the rationale of study, need of the study and purpose of the study. However, the research question is missing in the introduction section and the same can be incorporated.
We have included a new paragraph in the introduction: "In this sense, community pharmacists could help improve health by acting in certain aspects. On the one hand, they can decrease the probability of ADR and increase the reporting of low-severity events detected during dispensing. On the other hand, they can promote greater adherence to medication, strengthening and improving the quality of health care."
- Methodology needs further elaboration in order to make the things more clear and replicable.
Certainly, we have included a new information in this section: Inclusion and exclusion criteria and information of workshop. In this line, we have also added information on ADR in the results and discussion sections.
Thank you so much for all the interesting considerations.
Best regards.